# Multi-label Learning with Semantic Embeddings

**Liping Jing, MiaoMiao Cheng & Liu Yang**
Beijing Key Lab of Traffic Data Analysis and Mining
Beijing Jiaotong University
Beijing, China, 100044
`{lpjing,15112085,11112191}@bjtu.edu.cn`

**Alex Gittens & Michael W. Mahoney**
ICSI and Department of Statistics, University of California at Berkeley
Berkeley, CA, 94704
`gittens@icsi.berkeley.edu, mmahoney@stat.berkeley.edu`

## Abstract

Multi-label learning aims to automatically assign to an instance (e.g., an image or a document) the most relevant subset of labels from a large set of possible labels. The main challenge is to maintain accurate predictions while scaling efficiently on data sets with extremely large label sets and many training data points. We propose a simple but effective neural net approach, the **S**emantic **E**mbedding **M**odel (SEM), that models the labels for an instance as draws from a multinomial distribution parametrized by nonlinear functions of the instance features. A Gauss-Siedel mini-batch adaptive gradient descent algorithm is used to fit the model. To handle extremely large label sets, we propose and experimentally validate the efficacy of fitting randomly chosen marginal label distributions. Experimental results on eight real-world data sets show that SEM garners significant performance gains over existing methods. In particular, we compare SEM to four recent state-of-the-art algorithms (NNML, BMLPL, REmbed, and SLEEC) and find that SEM uniformly outperforms these algorithms in several widely used evaluation metrics while requiring significantly less training time.

## 1 Introduction

The multi-label learning problem is to learn to predict potentially multiple relevant labels given an instance. Instances that have multiple labels naturally occur in many application domains, including multimedia information retrieval, tag recommendation, semantic scene classification, query categorization, gene function prediction, medical diagnosis, drug discovery, and marketing.

A popular approach to the multi-label learning problem is to embed the labels in a low-dimensional latent space via linear or local non-linear embeddings. The approach of Hsu et al. (2009) projects the label vectors to a random low-dimensional space, fits a regression model in this space, then projects these predictions back to the original label space. Balasubramanian & Lebanon (2012) use a sparsity-regularized least squares reconstruction objective to select a small set of landmark labels that are used to predict the remaining labels. Bi & Kwok (2013) take a similar approach, with a greatly decreased computation cost, by posing the problem of selecting the landmark labels as one of column subset selection and adopting the leverage score sampling approach (Boutsidis et al., 2009). Recently, Yu et al. (2014) and Jing et al. (2015) propose using trace norm regularization to identify a low-dimensional representation of the original large label space. Mineiro & Karampatziakis (2015) use randomized dimensionality reduction to learn a low-dimensional embedding that explicitly captures correlations between the instance features and their labels. These approaches, like other linear embedding methods, assume that the label matrix is low-rank. However, the label matrix in most applications of multi-label learning is a sparse binary matrix, and thus is extremely likely to violate this low-rank assumption (Bhatia et al., 2015).

Rather than working with the original label and feature matrices, some methods work instead with label or feature similarity matrices, and seek to preserve the local structure of the data in the learned low-dimensional latent space. Tai & Lin (2010) use PCA on the label covariance matrix to extract a low-dimensional latent space for labels and Chen & Lin (2012) extend this method to integrate feature information. Lin et al. (2014) apply PCA to a similarity matrix constructed using both label and feature information; this approach is time-consuming as it requires computing a large similarity matrix. Nam et al. (2014) introduce a neural network model to capture non-linear relationships between the input features and the labels. However, this approach is computationally infeasible when the number of possible labels is large. Similarly, Cissé et al. (2016) shows that using a deep learning approach built on top of an informative partitioning of the label space gives good performance; the scalability of this method was not characterized. Prabhu & Varma (2014) propose a method to efficiently train a classification tree by minimizing the Normalized Discounted Cumulative Gain. Rai et al. (2015) assumes that the label vectors are generated by sampling from a weighted combination of label topics, where the mixture coefficients are determined by the instance features.

Bhatia et al. (2015) proposes a multi-phase algorithm (SLEEC) that first clusters the instances into a number of relatively small groups, learns label embeddings for each group via an SVD, and then trains linear regressors from the input features to the latent label factors for each group. SLEEC empirically outperforms previous state-of-the-art multi-label classifiers, but the label embedding in each group is learned from a nearest neighbor graph that is constructed solely from labelling information, ignoring the available feature matrix; the feature matrix has been shown repeatedly to be a source of useful information for label embedding (Chen & Lin, 2012; Lin et al., 2014; Yu et al., 2014; Jing et al., 2015).

The contribution of this paper is a scalable, accurate, and simple neural network approach to multi-label learning. Experiments establish that our method is faster and more accurate than SLEEC, the current state-of-the-art scalable algorithm.

**Notation:** In the sequel, $n$ is the number of training instances, $c$ is the cardinality of the set of possible labels, $d$ is the dimensionality of the feature vectors, and $r$ is the dimension of the learned latent space. The matrix $\mathbf{X} \in \mathbb{R}^{n \times d}$ contains the instance features, and $\mathbf{Y} \in {0, 1}^{n \times c}$ indicates the labels assigned to each instance. We denote the number of observed labels for instance $i$ with $\ell_i = \sum_{k=1}^{c} y_{ik}$. The notations $\mathbf{A}_{i\cdot}$ and $\mathbf{A}_{\cdot j}$ respectively refer to the $i$th row and $j$th column of the matrix $\mathbf{A}$. Unless otherwise specified, the notation $f(\mathbf{A})$ denotes the elementwise application of an arbitrary function $f$ to the $\mathbf{A}$, so for example $\exp(\mathbf{A})_{ij} = \exp(a_{ij})$.

## 2   THE SEMANTIC EMBEDDING MODEL

Our Semantic Embedding Model (SEM) assumes that the *underlying* parameters determining the observed labels are low-rank rather than that the observed label matrix is itself low-rank, and it uses a nonlinear model to fit the probability distributions over the labels, conditioned on the instance features.

SEM models the $i$-th row of $\mathbf{Y}$ as the result of $\ell_i$ draws from a multinomial distribution:

$$\mathbf{Y}_{i\cdot} \sim \text{Multinomial}(\ell_i; \mathbf{P}_i), \quad \text{where } \mathbf{P} = \left[ \frac{\exp(h_{ij})}{\sum_{k=1}^{c} \exp(h_{ik})} \right]_{\substack{i=1,\ldots,n \\ j=1,\ldots,c}}. \tag{1}$$

The parameter matrix $\mathbf{H} = \mathbf{U}\mathbf{V}^T + \mathbf{1}_n \mathbf{b}^T$ is the sum of label priors $\mathbf{b} \in \mathbb{R}^c$ and the product of explanatory latent factors associated with the instances ($\mathbf{U} \in \mathbb{R}^{n \times r}$) and the labels ($\mathbf{V} \in \mathbf{R}^{c \times r}$). Further, we allow the latent factors associated with each instance to be a nonlinear function of the features associated with that instance, $\mathbf{U} = f(\mathbf{X}, \mathbf{W})$ for some $\mathbf{W}$ to be learned. We note that if $f(\mathbf{X}, \mathbf{W}) = \mathbf{X}\mathbf{W}$, SEM could be viewed as fitting a Bayesian Exponential Family PCA (Mohamed et al., 2009). However, throughout this paper we take $f(\mathbf{X}\mathbf{W}) = \sigma(\mathbf{X}\mathbf{W})$, where $\sigma(\mathbf{X}) = (1 + \exp(-\mathbf{X}))^{-1}$ denotes the elementwise application of the sigmoid function, as we find this gives good results; with this choice, SEM is more naturally viewed as a neural network model.

We fit the SEM parameters by maximizing the likelihood of the observed labels. This is equivalent to minimizing the sum of the KL divergences between the empirical label distributions for each instance and the label distributions predicted by the model (Pawitan, 2001). Accordingly, we define the empirical label distribution matrix $\mathbf{G}$, whose $i$th row satisfies $\mathbf{G}_i = \mathbf{Y}_i/\ell_i$, then minimize the

row-wise Kullback-Leibler distance (Yang et al., 2011) between $\mathbf{G}$ and $\mathbf{P}$:

$$\mathcal{J}_{\mathbf{G}||\mathbf{P}} = \sum_{i=1}^{n} \sum_{j=1}^{c} \mathbf{G}_{ij} \log \frac{\mathbf{G}_{ij}}{\mathbf{P}_{ij}} \cong - \sum_{i=1}^{n} \sum_{j=1}^{c} \mathbf{G}_{ij} \log \mathbf{P}_{ij}. \tag{2}$$

Recalling that

$$\mathbf{P}_{ij} = \frac{\exp(h_{ij})}{\sum_{k=1}^{c} \exp(h_{ik})} = \frac{\exp\left((\sigma(\mathbf{XW})\mathbf{V}^T)_{ij} + b_j\right)}{\sum_{k=1}^{c} \exp\left((\sigma(\mathbf{XW})\mathbf{V}^T)_{ik} + b_k\right)},$$

some algebraic manipulations give the final objective

$$
\begin{aligned}
\mathcal{J}(\mathbf{W}, \mathbf{V}, \mathbf{b}) = \mathcal{J}_{\mathbf{G}||\mathbf{P}} &= - \sum_{i=1}^{n} \sum_{j=1}^{c} \mathbf{G}_{ij} \log \frac{\exp(\sigma(\mathbf{XW})_{i\cdot}(\mathbf{V}^T)_{\cdot j} + b_j)}{\sum_{k=1}^{c} \exp(\sigma(\mathbf{XW})_{i\cdot}(\mathbf{V}^T)_{\cdot k} + b_k)} \\
&= - \sum_{i=1}^{n} \sum_{j=1}^{c} \mathbf{G}_{ij}(\sigma(\mathbf{XW})_{i\cdot}(\mathbf{V}^T)_{\cdot j} + b_j) + \sum_{i=1}^{n} \log \left( \sum_{k=1}^{c} \exp(\sigma(\mathbf{XW})_{i\cdot}(\mathbf{V}^T)_{\cdot k} + b_k) \right) \\
&= -\mathrm{Tr}(\mathbf{G}(\sigma(\mathbf{XW})\mathbf{V}^T + \mathbf{1}_n \mathbf{b}^T)^T) + \mathbf{1}_n^T \log \left( \exp(\sigma(\mathbf{XW})\mathbf{V}^T + \mathbf{1}_n \mathbf{b}^T)\mathbf{1}_c \right).
\end{aligned}
\tag{3}
$$

Thus the SEM parameters are learned by solving the optimization problem

$$\min_{\mathbf{W}, \mathbf{V}, \mathbf{b}} \mathcal{J}(\mathbf{W}, \mathbf{V}, \mathbf{b}). \tag{4}$$

Here $\mathbf{V} \in \mathbb{R}^{c \times r}$ are the representations of the labels in a latent semantic space, $\mathbf{W} \in \mathbb{R}^{d \times r}$ controls the nonlinear mapping from the instance features to the same semantic space, and the offsets $\mathbf{b} \in \mathbb{R}^c$ allow for label-specific offsets in the mapping from the semantic space to the log probabilities.

## 3   MODEL FITTING

The optimization problem (4) is non-convex. To solve it efficiently, we use a Gauss-Siedel approach combined with mini-batching.

Namely, we cyclically update each of $\mathbf{W}, \mathbf{V},$ and $\mathbf{b}$ using AdaGrad (Duchi et al., 2011) while keeping the other two variable fixed. We compute the gradients using mini-batches. To state the expressions for the gradients with respect to the model parameters, we introduce some helpful notation: $\mathbf{A} \odot \mathbf{B}$ denotes the entry-wise product of two matrices, $\mathbf{M} = \sigma(\mathbf{XW}) \odot (1 - \sigma(\mathbf{XW}))$, and $\mathbf{D} = \mathrm{Diag}\left( \exp\left(\sigma(\mathbf{XW})\mathbf{V}^T + \mathbf{1}_n \mathbf{b}^T\right)\mathbf{1}_c \right)$. The gradients are readily computed from (2):

$$\mathcal{G}(\mathbf{W}) = \mathbf{X}^T \left( \mathbf{M} \odot \left[ \left( \mathbf{D}^{-1} \exp\left(\sigma(\mathbf{XW})\mathbf{V}^T + \mathbf{1}_n \mathbf{b}^T\right) - \mathbf{G} \right) \mathbf{V} \right] \right) \tag{5}$$

$$\mathcal{G}(\mathbf{V}) = \left( \exp\left(\sigma(\mathbf{XW})\mathbf{V}^T + \mathbf{1}_n \mathbf{b}^T\right)^T \mathbf{D}^{-1} - \mathbf{G}^T \right) \sigma(\mathbf{XW}) \tag{6}$$

$$\mathcal{G}(\mathbf{b}) = \left( \exp\left(\sigma(\mathbf{XW})\mathbf{V}^T + \mathbf{1}_n \mathbf{b}^T\right)^T \mathbf{D}^{-1} - \mathbf{G}^T \right) \mathbf{1}_n. \tag{7}$$

Using AdaGrad, the update rule for $\mathbf{W}^{(\tau)}$ is

$$\mathbf{W}^{(\tau)} = \mathbf{W}^{(\tau-1)} - \alpha_{\mathbf{W}}^{(\tau)} \odot \mathcal{G}(\mathbf{W}^{(\tau-1)}) \tag{8}$$

where $\tau$ is the timestep and $\alpha_{\mathbf{W}}$ is a matrix of step sizes computed via

$$\left(\alpha_{\mathbf{W}}^{(\tau)}\right)_{iq} = \frac{\rho}{\sqrt{\sum_{m=1}^{\tau-1} \left(\mathcal{G}(\mathbf{W}^{(m)}) \odot \mathcal{G}(\mathbf{W}^{(m)})\right)_{iq} + \varepsilon}}, \tag{9}$$

where $\varepsilon$ and the learning rate $\rho$ determine how much an entry $\mathbf{W}_{ij}$ is updated during the first timestep.

$\mathbf{V}^{(\tau)}$ and $\mathbf{b}^{(\tau)}$ are computed according to similar updating rules obtained from (8) and (9) by substituting $\mathcal{G}(\mathbf{W})$ with $\mathcal{G}(\mathbf{V})$ (or $\mathcal{G}(\mathbf{b})$), $\mathbf{W}$ with $\mathbf{V}$ (or $\mathbf{b}$), and $\alpha_{\mathbf{W}}$ with $\alpha_{\mathbf{V}}$ (or $\alpha_{\mathbf{b}}$).

A listing of the proposed algorithm is given in Algorithm 1. Its computational complexity is $O(Tnr(d+c))$, where $T$ is the number of epochs. We note that the gradient calculations in lines 7–9 of Algorithm 1 are amenable to parallelization.

---

**Algorithm 1** Mini-Batched Gauss-Siedel Adaptive Gradient Descent for learning SEM parameters

---

**Input:** Instance feature matrix $\mathbf{X} \in \mathbb{R}^{n \times d}$, observed label matrix $\mathbf{Y} \in \mathbb{R}^{n \times c}$, dimensionality of the latent space $r$, learning rate $\rho$ and $\epsilon > 0$, mini-batch size $m < n$, and number of epochs $T$.

1: Initialize $\mathbf{W}^{(0)}$, $\mathbf{V}^{(0)}$, and $\mathbf{b}^{(0)}$
2: **for** $t = 1, 2, \ldots, T$ **do**
3: Randomly choose $n/m$ mini-batchs $I_z \subset \{1, ..., n\}$ of size $m$
4: **for** $b = 1, 2, \ldots n/m$ **do**
5: Set $\tau = (t-1)(n/m) + b$
6: Select the data instances in the $z$-th mini-batch by working with $\mathbf{X}_{I_z, \cdot}$ in lieu of $\mathbf{X}$
7: Update $\mathbf{W}^{(\tau)}$ via (8) while fixing $\mathbf{V} = \mathbf{V}^{(\tau-1)}$ and $\mathbf{b} = \mathbf{b}^{(\tau-1)}$
8: Update $\mathbf{V}^{(\tau)}$ via the analog of (8) for $\mathbf{V}$ while fixing $\mathbf{W} = \mathbf{W}^{(\tau)}$ and $\mathbf{b} = \mathbf{b}^{(\tau-1)}$
9: Update $\mathbf{b}^{(\tau)}$ via the analog of (8) for $\mathbf{b}$ while fixing $\mathbf{W} = \mathbf{W}^{(\tau)}$ and $\mathbf{V} = \mathbf{V}^{(\tau)}$
10: **end for**
11: **end for**
**Output:** $\mathbf{W}^{(\tau)}$, $\mathbf{V}^{(\tau)}$ and $\mathbf{b}^{(\tau)}$

---

### 3.1 Increased efficiency by fitting marginals

Although Algorithm 1 runs in time linear in the dimensions of the model parameters and the input datasets, it can be computationally expensive when there are more than a few thousand labels. To further reduce the running time of our algorithm, we note that in practice, each instance is often associated with $\ell_i \ll c$ labels.

To speed up the training, at each timestep $\tau$, rather than attempting to minimize the divergence between the entire empirical and predicted label distributions of each instance, we sample a set of labels $L_i^{(\tau)}$ for each instance and attempt to minimize the empirical and predicted *marginal label distributions* over that set of labels $L_i^{(\tau)}$. Let $PL_i$ denote the set of labels assigned to the $i$th instance, and $AL_i$ denote the set of labels not assigned to that instance. We sample from $AL_i$ to form a set $NL_i$, and use $L_i^{(\tau)} = PL_i \cup NL_i^{(\tau)}$. This leads to the modified objective

$$\mathcal{J}_{\text{Marginal}}(\mathbf{W}, \mathbf{V}, \mathbf{b})^{(\tau)} = -\sum_{i=1}^{n} \sum_{j \in L_i^{(\tau)}} \mathbf{G}_{ij} \log \frac{\exp(\sigma(\mathbf{X}\mathbf{W})_{i\cdot}(\mathbf{V}^T)_{\cdot j} + b_j)}{\sum_{k \in L_i^{(\tau)}} \exp(\sigma(\mathbf{X}\mathbf{W})_{i\cdot}(\mathbf{V}^T)_{\cdot k} + b_k)}. \quad (10)$$

Note that $\mathcal{J}^{(\tau)}$ is a random function that changes at each timestep. Minimizing this stochastic objective effectively seeks SEM parameters which fit all the randomly sampled marginals encountered during training. Thus it is important to sample the sets $NL_i$ so that the selected marginals capture non-trivial information about the label distributions. One can imagine that uniformly sampling from $AL_i$ will not provide very informative marginals. As an improvement on this naïve scheme, we sample labels from $AL_i$ with probability proportional to their frequency of occurrence in the training data set. The number of negative labels is set to be $\beta$ times the number of positive labels i.e., $|NL_i| = \beta|PL_i| = \beta\ell_i$. Further, when $m > 1$, to faciliate efficient BLAS operations while mini-batching, we use the same marginals for each instance in the same minibatch, i.e., we fit marginals over $L^{(\tau)} := \bigcup_{i \in I_z} L_i^{(\tau)}$, where $I_z$ denotes the set of instances in the current minibatch.

In the experiments presented in Section 4, we found that $\beta$ around 10 suffices when $c$ is relatively small, and $\beta$ around 100 suffices when $c$ is on the order of tens of thousands.

### 3.2 Label Prediction

We present two methods for predicting the labels for a new instance $\mathbf{x} \in \mathbb{R}^d$ given the fitted SEM parameters.

The first uses the generative model behind SEM: form $\mathbf{h} = \sigma(\mathbf{x}^T \mathbf{W})\mathbf{V}^T + \mathbf{b}^T$ and note the probability that the $j$th label is assigned to that instance is given by

$$\mathbb{P}(y_j = 1) = \exp(h_j) / \sum_{k=1}^{c} \exp(h_k). \quad (11)$$

Accordingly we assign the most probable labels to $\mathbf{x}$. We call this prediction scheme the *direct SEM* method; it simply requires choosing the labels corresponding to the largest entries of $\mathbf{h}$.

The second method builds a kernel classifier in the semantic space obtained from the SEM factorization. Following Mineiro & Karampatziakis (2015), a classifier is trained on these semantic representations by solving the optimization problem

$$\min_{\mathbf{Z}\in\mathbb{R}^{c\times s}} \sum_{i=1}^{n} \ell(\mathbf{Y}_{i\cdot}, \mathbf{Z}\psi(\mathbf{x}_i\mathbf{W})) + \lambda\|\mathbf{Z}\|_F^2, \tag{12}$$

where $\ell$ is the log-loss penalty and $\psi : \mathbb{R}^r \to \mathbb{R}^s$ is an $s$-dimensional Random Fourier Feature (RFF) map (Rahimi & Recht, 2007):

$$\psi(\mathbf{x}) = \cos\left(\boldsymbol{\Phi}\mathbf{x} + \boldsymbol{\theta}\right), \tag{13}$$

where $\boldsymbol{\Phi} \in \mathbb{R}^{s\times r}$ is a matrix of i.i.d. standard Gaussians and $\boldsymbol{\theta} \in [0, 2\pi)^s$ is a vector of i.i.d uniform samples from $[0, 2\pi)$.

At test time, the predicted label probabilities for an instance $\mathbf{x}$ are given by $\mathbf{Z}\psi(\mathbf{x}\mathbf{W})$, so we assign the most probable labels according to this model. We refer to this scheme as the *kernelized SEM* method.

## 4 EXPERIMENTS

In the sequel we refer to the direct SEM scheme as simply SEM, and the kernelized SEM scheme as SEM-K. We compare SEM and SEM-K with several alternative multi-label learning algorithms: NNML (Nam et al., 2014), REmbed (Mineiro & Karampatziakis, 2015), SLEEC (Bhatia et al., 2015), and BMLPL (Rai et al., 2015). We do not compare to the models proposed in (Tai & Lin, 2010; Chen & Lin, 2012; Bi & Kwok, 2013; Yu et al., 2014; Prabhu & Varma, 2014) because earlier works (Yu et al., 2014; Bhatia et al., 2015) have shown that they are inferior to SLEEC.

### 4.1 DATASETS

Table 1 summarizes the eight datasets used in our experiments. Here $n_{\text{train}}$ and $n_{\text{test}}$ are the numbers of training and testing instances, $d$ is the number of features, $c$ is the number of labels/classes, and the avg($\ell_i$) column reports the average number of labels per instance. In these datasets, the number of labels varies from 23 to 30938, the average label cardinality varies from 2.508 to 19.020, and the number of instances in different classes varies over a large range. Thus predicting the labels assignments correctly over this collection of datasets is a challenging task.

Table 1: Multi-label dataset summary.

| Dataset | Domain | $n_{\text{train}}$ | $n_{\text{test}}$ | $d$ | $c$ | avg($\ell_i$) |
|---------|--------|--------|-------|------|------|--------|
| *MSRC* | image | 296 | 295 | 512 | 23 | 2.508 |
| *Corel5K* | image | 4500 | 500 | 499 | 374 | 3.522 |
| *SUN* | image | 12906 | 1434 | 512 | 102 | 15.526 |
| *Delicious* | text | 12920 | 3185 | 500 | 983 | 19.020 |
| *EurLex-sub* | text | 17413 | 1935 | 5000 | 201 | 2.213 |
| *Mediamill* | video | 30993 | 12914 | 210 | 101 | 4.736 |
| *Eurlex-des* | text | 17413 | 1935 | 5000 | 3993 | 5.31 |
| *Wiki10K* | text | 14146 | 6616 | 101938 | 30938 | 18.64 |

### 4.2 METHODOLOGY

The codes of the methods we compare to are provided by the authors, in particular, we note that the computationally intensive portions of REmbed, SLEEC and NNML are implemented in C; by way of comparison, our algorithms are entirely implemented in Matlab. Due to there being several parameters for each method, we hand-tuned the parameters for each dataset as suggested by the authors. All methods were run in MATLAB on a Windows server with 4GB memory and four 2.3GHz CPUs with eight cores.

The prediction performance for each algorithm is evaluated according to widely-used metrics in the field of multi-label classification, viz., label-based Macro-F1 (MaF1) and Micro-F1 (MiF1) and instance-based Precision-at-k (P@$k$, esp. P@1 and P@3) (Zhang & Zhou, 2014). MaF1 and MiF1 require predefining a threshold to determine the number of labels to be assigned to the testing data. In our experiments, the number of labels assigned to each testing instance was set according to its ground truth.

Table 2: The classification performance of six multi-label classification algorithms (NNML, BMLPL, REmbed, SLEEC and the proposed SEM and SEM-K). The best and second best results are respectively bolded and underlined for each evaluation measure.

| | MaF1 | MiF1 | P@1 | P@3 | MaF1 | MiF1 | P@1 | P@3 |
|---|---|---|---|---|---|---|---|---|
| | *MSRC* | | | | *Corel5K* | | | |
| NNML | 0.4086 | 0.5944 | 0.7356 | 0.5073 | 0.0547 | 0.2967 | 0.4020 | 0.3047 |
| BMLPL | 0.4592 | 0.6199 | 0.7017 | 0.5288 | 0.0315 | 0.2779 | 0.3940 | 0.2820 |
| REmbed | 0.3537 | 0.5128 | 0.5322 | 0.4384 | 0.0450 | 0.2144 | 0.3060 | 0.2247 |
| SLEEC | 0.4973 | 0.6314 | 0.7353 | 0.5243 | 0.0534 | **0.3188** | **0.4360** | 0.3287 |
| SEM | 0.5064 | 0.6173 | 0.7220 | 0.5333 | **0.0623** | **0.3188** | 0.4320 | **0.3293** |
| SEM-K | **0.5770** | **0.6492** | **0.7458** | **0.5525** | 0.0589 | 0.2649 | 0.3600 | 0.2773 |
| | *SUN* | | | | *Mediamill* | | | |
| NNML | 0.2807 | 0.5248 | 0.9421 | 0.8580 | 0.0819 | 0.5890 | 0.8260 | 0.6675 |
| BMLPL | 0.1897 | 0.4766 | 0.9024 | 0.8001 | 0.0855 | 0.6012 | 0.8478 | 0.6854 |
| REmbed | 0.3408 | 0.5125 | 0.9393 | 0.8591 | 0.2634 | 0.6371 | 0.8741 | 0.6988 |
| SLEEC | 0.2935 | 0.5256 | 0.9484 | 0.8656 | **0.2851** | 0.6546 | 0.8899 | 0.7158 |
| SEM | 0.3648 | **0.5486** | 0.9365 | 0.8642 | 0.1593 | 0.6296 | 0.8746 | 0.6996 |
| SEM-K | **0.3703** | 0.5466 | **0.9575** | **0.8787** | 0.2570 | **0.6717** | **0.8953** | **0.7278** |
| | *Delicious* | | | | *Eurlex-sub* | | | |
| NNML | 0.1721 | 0.3963 | 0.6687 | **0.6169** | 0.5761 | 0.8487 | 0.9173 | 0.6267 |
| BMLPL | 0.1061 | 0.3739 | 0.6378 | 0.5772 | 0.1459 | 0.6011 | 0.6789 | 0.4697 |
| REmbed | 0.1549 | 0.3713 | 0.6353 | 0.572 | 0.5335 | 0.8031 | 0.8785 | 0.5977 |
| SLEEC | 0.1257 | 0.3859 | 0.6674 | 0.6112 | 0.5433 | 0.8461 | 0.9152 | 0.6191 |
| SEM | **0.1941** | **0.3980** | **0.6727** | 0.6162 | 0.5652 | 0.8339 | 0.8971 | 0.6188 |
| SEM-K | 0.1675 | 0.3886 | 0.6658 | 0.6112 | **0.5807** | **0.8494** | **0.9188** | **0.6269** |

Table 3: The running times, in seconds, of six multi-label classification algorithms (NNML, BMLPL, REmbed, SLEEC and the proposed SEM and SEM-K) for differing training sizes on the *Mediamill* dataset.

| $n_{train}$ | NNML | BMLPL | REMBED | SLEEC | SEM | SEM-K |
|---|---|---|---|---|---|---|
| 439 | 327.57 | 10.29 | 2.07 | 16.11 | 0.60 | 1.50 |
| 1756 | 1333.91 | 20.35 | 3.02 | 57.16 | 2.41 | 4.29 |
| 3073 | 2363.02 | 48.2 | 4.14 | 145.88 | 4.36 | 6.99 |
| 4391 | 3264.79 | 41.72 | 5.45 | 227.76 | 6.65 | 10.10 |
| 8781 | 4428.09 | 84.09 | 10.83 | 815.66 | 12.29 | 21.73 |
| 13172 | 5170.00 | 119.09 | 17.04 | 1041.07 | 18.39 | 26.49 |
| 17563 | 5170.17 | 185.05 | 20.90 | 1692.7 | 24.22 | 42.21 |
| 21954 | 5297.75 | 225.96 | 44.20 | 1772.52 | 30.10 | 50.64 |
| 26344 | 5947.94 | 235.93 | 52.75 | 1985.82 | 35.95 | 59.42 |
| 30735 | 6604.93 | 275.06 | 58.74 | 2181.48 | 41.37 | 61.30 |

## 4.3 PERFORMANCE ON DATASETS WITH SMALL LABEL SETS

First we compare the performance on six multi-label learning problems with $c < 1000$. To fit both SEM models, we take the number of epochs be 30 and the mini-batch size be 200—i.e., $T = 30$ and $m = 200$ in Algorithm 1—and because $c$ is small, we fit the full label distributions. The classification performances of our SEM algorithms and the baseline methods are shown in Table 2. SEM or SEM-K outperform the alternative algorithms in most cases.

Table 3 compares the running times of the algorithms as the size of the dataset is increased, using *MediaMill*. We see that SEM is the fastest model, followed by REMBED, then closely by SEM-K; the remaining three models are significantly more costly. It is clear that NNML, the previous neural

network approach to multi-label learning costs the most. In the other five algorithms, the latent space dimensionality ($r$) is set to be 50. SLEEC is expensive because it constructs the nearest neighbor graph among training data and computes the top $r$ eigenvectors of the corresponding similarity matrix, which costs $O(n^2r + d^2r)$. REmbed is efficient because its main cost is to find the singular vectors of a $c \times (r + q)$ matrix (here $c$ is the number of labels and $q$ is a small integer), but its performance is inferior to SEM-K. The BMLPL code provided by the author applies SVD to the training data to initialize than model parameters and then uses conjugate gradient to update the parameters, thus it costs much more than REmbed and our proposed methods.

## 4.4 PERFORMANCE ON DATASETS WITH LARGE LABEL SETS

We proposed using SEM to fit marginals rather than the entire label distribution when $c$ is large, for computational efficiency. To judge the effectiveness of this proposal, we compare the accuracy and running times of the SEM and SEM-K models with baselines on *EurLex-des* and *Wiki10K*, two datasets with $c > 1000$. As baselines, we use REmbed and SLEEC in accordance with the above discussion which showed that these two methods are efficient and/or have good performance.

The hyperparameters in SLEEC were set according to the original authors' code: $r$ for *EurLex-des* and *Wiki10K* is 100 and 75 respectively, and 3 clusters are used for *Eurlex-des* and 5 are used for *Wiki10K*. To fit the SEM models, we used the same value of $r$ as SLEEC on these two datasets and used 10 training epochs. For REmbed, the latent space size $r$ was tuned via cross-validation; $r = 300$ for *Eurlex-des* and $r = 150$ for *Wiki10K*. The number of Random Fourier Features is 2000 for both REmbed and SEM-K. The latent space size $r$ in SEM is same with SLEEC. The mini-batch sizes and number of epochs are set to be 200 and 10 respectively when fitting the SEM models. The number of threads is set to be 8 for all methods.

Table 4: The classification performance of five methods (REmbed, SLEEC and the proposed SEM and SEM-K with two values of $\beta$) on the *Eurlex-des* and *Wiki10K* datasets. The best and second best results are respectively bolded and underlined for each evaluation metric.

| | | REmbed | SLEEC | SEM ($\beta = 500$) | SEM-K ($\beta = 10$) | SEM-K ($\beta = 60$) |
|---|---|---|---|---|---|---|
| | P@1 | 0.7299 | 0.8017 | 0.7107 | 0.8024 | **0.8135** |
| *Eurlex-des* | P@3 | 0.6064 | 0.6539 | 0.5874 | 0.6621 | **0.6714** |
| | P@5 | 0.5060 | 0.5375 | 0.4916 | 0.5493 | **0.5563** |
| | | REmbed | SLEEC | SEM ($\beta = 1200$) | SEM-K ($\beta = 10$) | SEM-K ($\beta = 100$) |
| | P@1 | 0.6963 | 0.8554 | 0.8517 | 0.8582 | **0.8671** |
| *Wiki10K* | P@3 | 0.5790 | 0.7359 | 0.7133 | 0.7278 | **0.7385** |
| | P@5 | 0.4929 | 0.6310 | 0.6171 | 0.6236 | **0.6353** |

Table 5: The running times, in seconds, of five methods (REmbed, SLEEC and the proposed SEM and SEM-K for two values of $\beta$) on the *Eurlex-des* and *Wiki10K* datasets.

| | REmbed | SLEEC | SEM ($\beta = 500$) | SEM-K ($\beta = 10$) | SEM-K ($\beta = 60$) |
|---|---|---|---|---|---|
| *Eurlex-des* | 358.63 | 1571.30 | 1210.30 | 167.10 | 250.77 |
| *Wiki10K* | 2858.96 | 2497.00 | 2003.43 | 646.48 | 769.18 |

Table 4 compares the classification performances of the methods on these two datasets. It is clear that SEM-K with a small set of negative labels obtains better performance than both REmbed and SLEEC. Table 5 shows that, additionally, the SEM-K models are fit much faster than than the other models.

## 4.5 IMPACT OF THE SIZE OF THE MARGINALS

Figure 1 illustrates the impact of the choice of $\beta$ on the prediction performance (in terms of P@1) of SEM and SEM-K. The performances of SLEEC and REmbed are included for comparison. The hyperparameters of SLEEC, REmbed and SEM were set as in Section 4.4.

It is evident that the performance of SEM increases significantly in a monotonic fashion with $\beta$. However, SEM-K is insensitive to $\beta$ once it passes a dataset-dependent threshold (e.g., $\beta = 60$

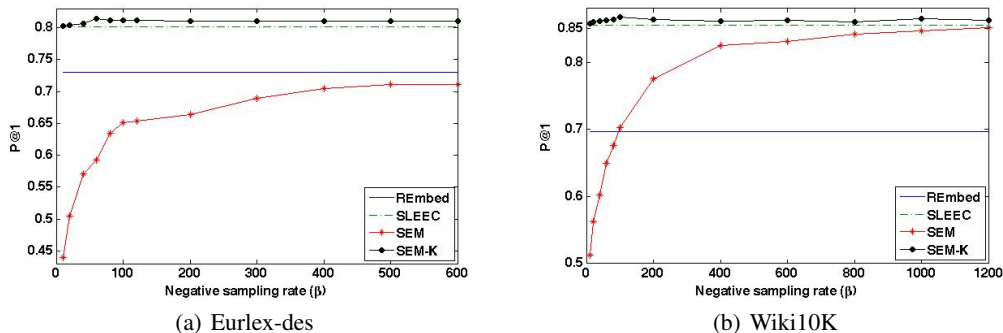

|                |                |
| :------------: | :------------: |
| (a) Eurlex-des | (b) Wiki10K   |

Figure 1: The P@1 performance of SEM and SEM-K as a function of $\beta$, in comparison to the performances of SLEEC and REmbed on the (a) *Eurlex-des* and (b)*Wiki10K* datasets.

for *Eurlex-des* and $\beta = 100$ for *Wiki10K*). Note that on *Wiki10K*, even the simpler direct SEM outperforms REmbed when there are sufficient negative labels.

Figure 2 illustrates the effect of $\beta$ on the running times of SEM and SEM-K. Note that the additional time to fit the classifier in the semantic space required by SEM-K is negligible compared to the time it takes to first fit the direct SEM model.

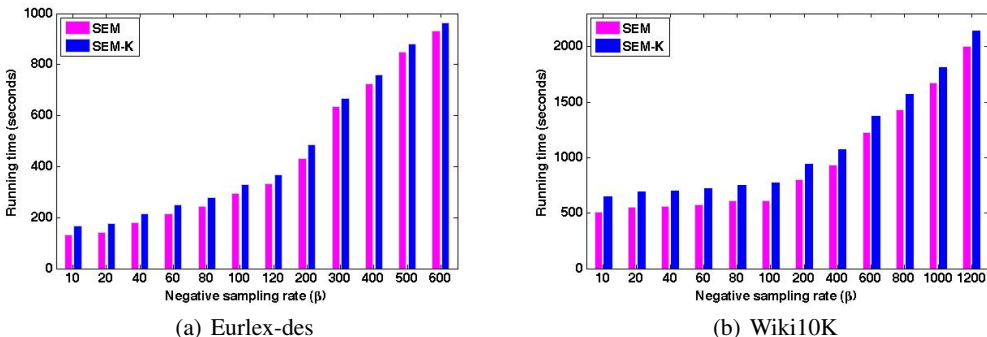

|                |                |
| :------------: | :------------: |
| (a) Eurlex-des | (b) Wiki10K   |

Figure 2: Running time of SEM-K under varying $\beta$.

### 4.6 ADDITIONAL CONSIDERATIONS

There are other important ways in which the proposed SEM methods can be compared to the baseline multi-label learning methods, including their performance as a function of the latent space dimensionality and as a function of the amount of training. Due to space constraints, a discussion of these two concerns and the convergence behavior of Algorithm 1 is provided in the Supplementary material.

## 5 CONCLUSION

We proposed a new semantic embedding model (SEM) for handling the multi-label learning task. A framework based on Gauss-Siedel mini-batched adaptive gradient descent was proposed for efficiently solving the non-convex optimization problem required to learn the SEM parameters. For large label sets, we proposed fitting the SEM to marginal distributions rather than the full label distribution. A series of experiments on eight real-world datasets empirically demonstrated that the proposed method is superior to state-of-the-art methods in terms of prediction performance and running time.

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

SUPPLEMENTARY MATERIAL

## A. Effect of Latent Space Dimensionality

It can be seen that the latent space dimensionality $r$ plays an important role to learn latent factors $\mathbf{V}$ and a feature mapping matrix $\mathbf{W}$ in our proposed methods, as it does in the three baselines BMLPL, REmbed and SLEEC. In order to investigate this dependence, we conducted a series of experiments on the training data sets using 5-fold cross-validation, comparing BMLPL, REmbed, SLEEC and our proposed SEM and SEM-K.

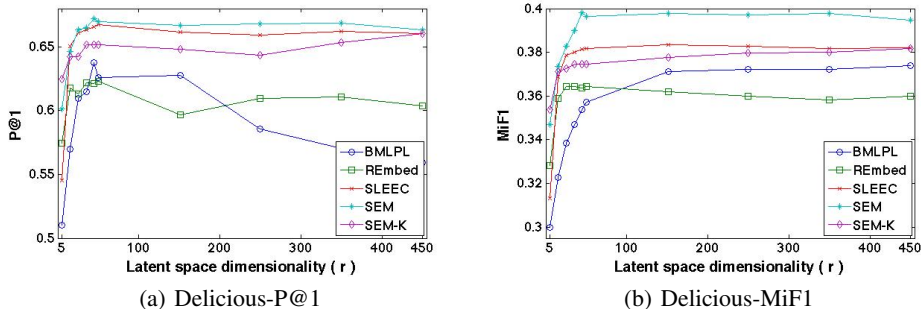

(a) Delicious-P@1 (b) Delicious-MiF1

Figure 3: The effect of the latent space dimensionality $r$ on BMLPL, REmbed, SLEEC, SEM and SEM-K in terms of MiF1 and P@1 on the *Delicious* dataset.

In this experiment, we take *Delicious* dataset as an example. The training data is separated into five folds where four folds are used as training and one fold as validating, and the averaged results in terms of P@1 and MiF1 are given by Figure 3. It can be seen that their performances usually improve with increasing $r$ until they reach an optimum value. However, once $r$ becomes too large, their performances degrade. This is reasonable: when $r$ is too small, the learned parameters cannot fully characterize the hidden semantic structure in the classification problem, while when $r$ is too large, the benefits of dimensionality reduction are lost, as the model begins to over-fit to the idiosyncrasies of the training data rather than capturing the semantic structure common to both the training and validation data. Usually, these methods could obtain good performance at small $r$, say 45 for *Delicious* dataset.

## B. Effect of training data size

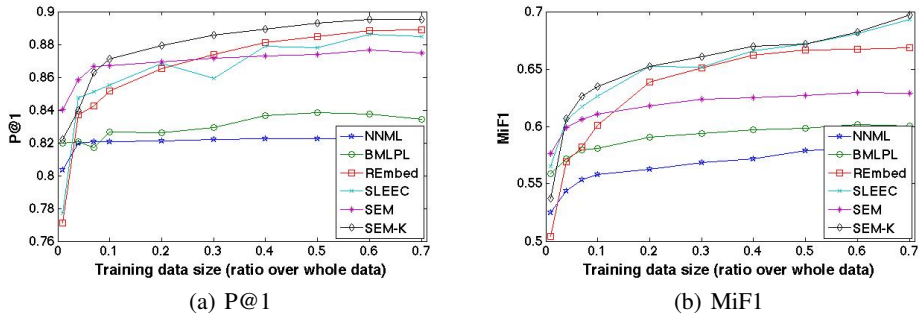

(a) P@1 (b) MiF1

Figure 4: Effect of varying the training data size, as a fraction of the combined test and training data, on five multi-label learning methods in terms of P@1 and MiF1 on the *Mediamill* dataset.

Meanwhile, we studied the label prediction performance as a function of the amount of labeled training data. In this experiment, we fixed the testing data size, and randomly selected training data from the training set so that the training data size varies from 1% to 70% of the combined training and testing data. In order to avoid the presence of empty categories and instances with no labels, at least one instance is kept for each label and at least one label is kept for each instance during this sampling process. For each fixed size of the training set, the desired amount of data is randomly

sampled ten times, and the resulting average P@1 and MiF1 on the testing data are recorded. During training, the latent dimensionality parameter $r$ is selected via 5-fold cross-validation.

Figure 4 shows these results for the Mediamill dataset which contains the largest number of instances. As expected, the performance of all the methods is positive correlated with the size of the training data set, and we also see that the proposed SEM-K uniformly outperforms the other methods regardless of the training data size. As it is often expensive to obtain large labeled data sets in real applications, this observation suggests that SEM-K is a better choice for these situations.

## C. Convergence

In order to demonstrated the convergence of the proposed method, we show the value of objective function (4) (at $r = 45$) via Figure 5(a) and the prediction result (P@1) via Figure 5(b) along with the number of passes to the dataset (i.e., $\tau$ in Algorithm 1). It can be seen that SEM could be convergent and the prediction performance becomes stable in less than 50 epochs, which will leverage SEM dealing with large-scale data.

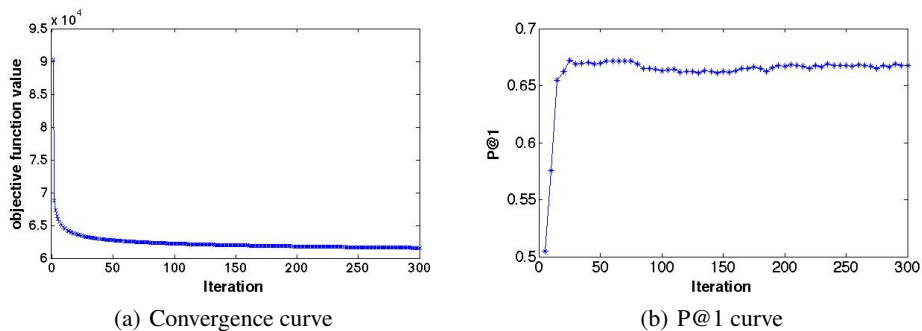

(a) Convergence curve      (b) P@1 curve

Figure 5: Performance of the proposed SEM method (with $r = 45, \rho = 0.1$) on the *Delicious* dataset, a) objective function value to minimum and b) prediction result in terms of P@1, where x-axis represents the number of passes to the dataset.

