# Peer review of "Multi-label learning with semantic embeddings"

_ICLR 2017 — rejected_

[Official Review · AnonReviewer2 · rating 4 · confidence 4 · 15 Dec 2016]

The paper presents the semantic embedding model for multi-label prediction.
In my questions, I pointed that the proposed approach assumes the number of labels to predict is known, and the authors said this was an orthogonal question, although I don't think it is!
I was trying to understand how different is SEM from a basic MLP with softmax output which would be trained with a two step approach instead of stochastic gradient descent. It seems reasonable given their similarity to compare to this very basic baseline.
Regarding the sampling strategy to estimate the posterior distribution, and the difference with Jean et al, I agree it is slightly different but I think you should definitely refer to it and point to the differences.
One last question: why is it called "semantic" embeddings? usually this term is used to show some semantic meaning between trained embeddings, but this doesn't seem to appear in this paper.

[Official Review · AnonReviewer1 · rating 5 · confidence 4 · 15 Dec 2016]
**Weak reject**

This paper proposes SEM, a simple large-size multilabel learning algorithm which models the probability of each label as softmax(sigmoid(W^T X) + b), so a one-layer hidden network. This in and of itself is not novel, nor is the idea of optimizing this by adagrad. Though it's weird that the paper explicitly derives the gradient and suggests doing alternating adagrad steps instead of the more standard adagrad steps; it's unclear whether this matters at all for performance. The main trick responsible for increasing the efficiency of this model is the candidate label sampling, which is done in a relatively standard way by sampling labels proportionally to their frequency in the dataset.

Given that neither the model nor the training strategy is novel, it's surprising that the results are better than the state-of-the-art in quality and efficiency (though non-asymptotic efficiency claims are always questionable since implementation effort trades off fairly well against performance). I feel like this paper doesn't quite meet the bar.

[Official Review · AnonReviewer3 · rating 4 · confidence 4 · 20 Dec 2016]
**not very convinced**

The paper proposes a semantic embedding based approach to multilabel classification. 
Conversely to previous proposals, SEM considers the underlying parameters determining the
observed labels are low-rank rather than that the observed label matrix is itself low-rank. 
However, It is not clear to what extent the difference between the two assumptions is significant

SEM models the labels for an instance as draws from a multinomial distribution
parametrized by nonlinear functions of the instance features. As such, it is a neural network.
The proposed training algorithm is slightly more complicated than vanilla backprop.  The significance of the results compared to NNML (in particular on large datasets Delicious and EUrlex) is not very clear. 

The paper is well written and the main idea is clearly presented. However, the experimental results are not significant enough to compensate the lack of conceptual novelty.

[Final Decision · Program Chairs · 06 Feb 2017]
**ICLR committee final decision**

This is largely a well written paper proposing a sensible approach for multilabel learning that is shown to be effective in practice. However, the main technical elements of this work: the model used and its connections to basic MLPs and related methods in the literature, the optimization strategy, and the speedup tricks are all familiar from prior work. Hence the reviewers are somewhat unanimous in their view that the novelty aspect of this paper is its main shortcomings. The authors are encouraged to revise the paper and clarify the precise contributions.